# Segregated by Wealth, Health, and Development: An Analysis of Pre-School Child Health in a Medium-Sized German City

**DOI:** 10.3390/children10050865

**Published:** 2023-05-12

**Authors:** Karoline Wagner, Andreas Wienke, Christine Gröger, Jan-Henning Klusmann, Amand Führer

**Affiliations:** 1Institute for Medical Epidemiology, Biometrics and Informatics (IMEBI), Interdisciplinary Center for Health Sciences, Medical School of the Martin Luther University Halle-Wittenberg, 06112 Halle, Germany; 2Public Health Department, 06110 Halle, Germany; 3Department of Pediatrics, Goethe University Frankfurt, 60590 Frankfurt, Germany

**Keywords:** child health, child, preschool, health status disparities, socioeconomic factors, Germany, Social Determinants of Health, Health Inequities

## Abstract

The School Entry Examination (SEE) can be used to identify children with current health issues, developmental delays, and risk factors for later diseases. This study analyzes the health status of preschool children in a German city with considerable socio-economic differences among its quarters. We used secondary data from SEEs 2016–2019 from the entire city (8417 children), which we divided into quarters with low (LSEB), medium (MSEB), and high socioeconomic burden (HSEB). In HSEB quarters, 11.3% of children were overweight as opposed to 5.3% in LSEB quarters. In HSEB quarters, 17.2% of children had sub-par cognitive development in contrast to 1.5% in LSEB quarters. For overall sub-par development, LSEB quarters had a prevalence of 3.3%, whereas, in HSEB quarters, 35.8% of children received this result. Logistic regression was used to determine the influence of the city quarter on the outcome of overall sub-par development. Here, considerable disparities among HSEB and LSEB quarters remained after adjustment for parents’ employment status and education. Pre-school children in HSEB quarters showed a higher risk for later disease than children in LSEB quarters. The city quarter had an association with child health and development that should be considered in the formulation of interventions.

## 1. Introduction

The foundations for socioeconomic inequality and later-life disease are established as early as in utero. If the expectant mother lives in deprived circumstances, this can lead to children being born prematurely or with low birthweight, which are both risk factors for diseases in later life [1,2,3]. Growing up in poverty is associated with poor physical and mental health [4,5], as well as impaired cognitive development [6] and poor educational achievement [7]. Even if socioeconomic circumstances improve in later life, a higher risk for disease remains, especially for adult-onset Type II diabetes, cardiovascular disease, and stroke [4]. Chronic degenerative diseases such as these are among the most common causes of disability and premature death [8,9].

Despite these potentially lifelong consequences, childhood poverty remains an issue even in wealthy countries. In Germany, which is the biggest economy in the EU, approximately 2.8 million children live in households with an income of less than 60% of the national average, which means that approximately every fifth child is at risk of poverty [10]. Rather than showing signs of improvement, childhood poverty has actually slightly increased since 2011 [11].

Some areas of the country even exceed this proportion. The city of Halle is located in the Eastern German state of Saxony-Anhalt and has a population of approximately 233,500 inhabitants [12]. It has an overall child poverty level of 33%, which is currently the fourth highest among major cities in Germany [13]. Within this city, some quarters far exceed this, with child poverty levels of approximately 70% in at least two quarters and approximately 50% in another three. The city has therefore been considered to be socially and economically segregated [14].

Living in deprived neighborhoods in Germany has been associated with increased morbidity [15]. In the city of Hamburg, the mean age at death differs by up to sixteen years between wealthy and poor neighborhoods [15,16]. However, the effects of living in deprived areas manifest even at much earlier ages. Children living in deprived neighborhoods, for instance, have a higher prevalence of asthma [17] and obesity [18].

Previous research has also shown throughout the world that living in a deprived neighborhood negatively influences health and health behaviors [19,20,21]. Some researchers have found this association regardless of individual financial circumstances, meaning that residence in a deprived quarter is in itself harmful to health [19].

### 1.1. The School Entry Examination

Children in Germany undergo a number of mandatory health check-ups before and during their schooling. One of these is the School Entry Examination (SEE), an examination undergone by all children approximately a year before starting school. The examination includes several standardized developmental tests, which assess a child’s developmental status with respect to fine and gross motor skills, language, and cognition. In addition, there are tests for vision, hearing, and general physical health. The results provide an overview of the current state of health and highlight possible needs for early intervention before starting school [20]. Data are also collected on the children’s place of residence, so it is known in which quarter of the city they live.

Previous research has shown that the SEE can be a useful tool to identify children at higher risk of health issues and developmental delays [21,22,23].

### 1.2. Aim of the Study

At present, there is a lack of research investigating the influence of high levels of socioeconomic deprivation within some quarters of European cities on the health and developmental status of preschoolers. Furthermore, few studies consider the effects of de facto socioeconomic segregation within Europe on these outcomes.

Using SEE data from Halle with its overall high level of economic deprivation and high levels of socioeconomic segregation within the city, we aim to investigate whether the city quarter constitutes—independently from individual socioeconomic status—a risk factor for poor medical and developmental outcomes, with a focus on the prevalence of established risk factors for chronic diseases in later life. This in turn provides an indication of the city quarters’ contextual influence on future health and professional trajectories.

## 2. Materials and Methods

### 2.1. Data

This is a retrospective study involving anonymized secondary data concerning preschoolers. The data were collected by the local Public Health Department as part of their work routine. The SEE is a mandatory examination for all prospective school children, and its results are frequently used for research purposes. Ethics approval was obtained from the ethics committee of the Medical Faculty Halle-Wittenberg (2019-170).

The data were collected by the public health department as part of their routine work and exported from the public health department’s administrative software. The data were exported in an already-anonymized form and transferred to the authors as a CSV database.

We included the annual cohorts from 2016 until 2019 of preschoolers examined by the Public Health Department in the city of Halle, which provided a dataset of 8417 children across the entire city.

For each child, the dataset comprises information from a clinical examination and developmental tests as well as information collected via a questionnaire on family circumstances completed by the parent or guardian prior to the SEE. This provides some information on the family’s demographic characteristics, such as parents’ marital status, the number of children in the household, and some data regarding parents’ education and employment status. Furthermore, parents or guardians are asked to bring their child’s health booklet (the so-called *U-Heft*), in which pediatricians document the child’s physical status at birth such as birth weight and gestational age, as well as the results of previous pediatric check-ups and diagnoses. Relevant results from the *U-Heft* were included in the data from the SEE and were therefore available for our analysis. If parents did not bring this booklet with them to the SEE, the results were not available.

In combination with the examination at the SEE, this allows insight into the trajectory of the child’s health and development from birth until approximately the age of five. Thus, the SEE provides an overall picture of child health and development.

### 2.2. Variables

The SEE provides extensive information on each child, not all of which have predictive value for later diseases or developmental delays. The full SEE comprises more than 200 variables. Of those, we included variables from three categories for our research. Variables were selected by clinical relevance but also included risk factors identified by earlier authors.

The first category included known risk factors for later physical diseases either identified in the clinical examination during the SEE or from the information provided in the *U-Heft*. These included current overweight as classified according to Kromeyer-Hauschild et al. [24], pre-term birth (before completing 37 weeks of gestation), and low birth weight (below 2500 g). All of these variables have been identified as influencing the long-term risks for diabetes and cardiovascular disease [25,26].

The second category provides an overview of the child’s current state of development: This includes standardized developmental tests that assess grammar, articulation, fine and gross motor skills, and overall cognitive development with reference to the child’s age [20]. These tests are conducted and evaluated by a public health department physician. Examples of skills tested include coordination, movement, spatial reasoning, and comparison of different items. We used a dichotomized variable, which distinguished between an overall result of sub-par development (below the cut-off score) and unremarkable development (above the cut-off score). These variables also hold significance for a child’s future success at school.

The third category of variables indicated the child’s usage of health care and social institutions. Here, we classified children’s participation in the last (voluntary) pediatric check-up preceding the SEE (U8 examination) as a dichotomous proxy for the child’s use of regular pediatric care (participated in check-up vs. did not participate). It also included kindergarten enrolment as a dichotomous indicator of the child’s social environment. We also looked at the proportion of new referrals to a specialist for a sub-par developmental result within the course of the SEE. These included speech therapy and occupational therapy as separate variables. These referrals were used as a second variable to determine the quality of the child’s regular pediatric care, as it can be expected that a child with good access to a pediatrician would already have received this diagnosis at an earlier point in time [27].

Maps showing the prevalence of variables from these three categories across the city were created in R.

Information on family circumstances includes the Brandenburger Sozialindex (BS) [28], which is divided into three categories of low, medium, and high BS, respectively. The BS takes into account the years of schooling completed by each parent and whether the parents are currently working. However, it does not distinguish between part-time and full-time employment, nor is there any information on income or tertiary education. The BS distinguishes between high levels of education of more than 10 years of formal education (three points), an average level of education of 10 years of schooling (two points), and a low level of education of less than 10 years (one point). For employment, it simply distinguishes between employment, irrespective of whether this is full-time or part-time (two points), and not employed (one point). Points are calculated for each parent individually and then added together to form the BS. Thus, the highest possible score is ten and the lowest possible score is four points. In the case of single-parent households, the single-parent score is doubled to reflect a hypothetical two-parent household. High BS is a score of nine or ten points, medium BS is seven or eight points, and low BS is four to six points.

In our data, we had information regarding the three categories of the BS, as well as its four separate components (mother’s and father’s individual level of education and employment status).

This study uses the term quarter to refer to the communal administrative units in the city of Halle. There are a total of 44 quarters. We excluded city quarters with less than 30 children from the analysis, which left 34 quarters. Previous researchers have divided Halle’s quarters according to their socioeconomic burden into quarters of low, medium, and high socioeconomic burden [13]. Factors considered for this classification included long-term unemployment, unemployment amongst young adults, numbers of households dependent on government subsidies, dependency on government subsidies despite being employed, and childhood poverty. The authors used a stepped index calculation to determine the quarters’ socioeconomic burden. For this project, we refer to their classification of Halle’s quarters into low, medium, and high socioeconomic burdens.

### 2.3. Statistical Analyses

The statistical analysis was performed in several steps:

First, we used descriptive statistics to determine the prevalence of risk factors and developmental issues for the whole city and for each quarter, as well as to describe the basic demographic characteristics of the study population.

Secondly, we first performed a simple logistic regression for the association between the quarter (independent variable) and sub-par development (dependent variable). In the next model (model A), we adjusted for individual BS as a proxy for education and employment. Subsequently, we adjusted for the four individual components of the BS in a further model (model B): Mother’s education, mother’s employment, father’s education, and father’s employment. As it is possible that the BS overestimates the social status of single-parent families by simply doubling the single-parent score, this method has the potential to yield more differentiated results.

## 3. Results

### 3.1. Sociodemographic Characteristics of the Study Population

The study population was evenly distributed with regard to sex. The mean age at the time of the SEE was 5 years, with small variability across the city’s quarters. However, there are considerable differences among the quarters regarding socioeconomic characteristics such as BS and family living arrangements. In quarters with low socioeconomic burden, there were less than 2% of children with a low BS, while over half of the children had a high BS. This is in stark contrast to the quarters with high socioeconomic burden, where the majority of children had a low BS. Table 1 shows the sociodemographic characteristics of our study population.

Furthermore, there were merely 8.9% (n = 48) of children living in single-parent households in the quarters with low socioeconomic burden, whereas, in the quarters with high socioeconomic burden, 32.4% (n = 869) lived with only one parent, mostly their mothers (30.4%; n = 816).

Children’s countries of origin also varied among the quarters. In quarters with low socioeconomic burden, 96.7% of children were born in Germany, in contrast to 75.7% in quarters with high socioeconomic burden.

There were also differences among the quarters regarding families’ health-related behaviors. In the quarters with low socioeconomic burden, the vast majority of children lived in non-smoking households, whereas in quarters with high socioeconomic burden, it was merely 41.7%.

### 3.2. Health Disparities across the Quarters

The descriptive analysis shows that among the different quarters of Halle, there are considerable differences in the prevalence of physical, social, and developmental risk factors, as well as differences in the utilization of healthcare. For each of the three categories, one variable was chosen for visual presentation in a choropleth map, showing its prevalence across all quarters of the city of Halle (Figure 1, Figure 2 and Figure 3).

#### 3.2.1. Developmental Tests

In quarters with high socioeconomic burden, 35.6% of all children scored below the cut-off value in grammar tests, whereas only 1.8% of children in quarters with low socioeconomic burden received this result. Overall, 17.4% of children received this result in the city of Halle.

With 35.8%, quarters with high socioeconomic burden had the highest proportion of children whose overall score in the developmental tests fell below the standard for normal development. This is in stark contrast to merely 3.5% in quarters with low socioeconomic burden and 17.7% in the entire cohort. Conversely, there were only 16.2% new referrals to a specialist for children living in quarters with high socioeconomic burden, which is less than half the proportion of sub-par development tests. In quarters with low socioeconomic burden, new referrals (7.8%) were double the proportion of sub-par developmental tests. Figure 1 shows the prevalence of an overall sub-par score in the developmental tests.

#### 3.2.2. Physical Risk Factors

In the category of physical risk factors, the differences between the quarters and the city of Halle are less pronounced. However, quarters with high socioeconomic burden still had more than twice the prevalence of overweight than quarters with low socioeconomic burden (11.3% vs. 5.3%).

Quarters with high socioeconomic burden have considerably more children born prematurely (11.2%) and underweight (9.9%) than quarters with low socioeconomic burden (prematurity 6.9%, low birthweight 4.9%), and still a considerably higher prevalence than the cohort average in Halle (prematurity 9.4%, low birthweight 7.3%). Figure 2 shows the prevalence of overweight across the city of Halle.

#### 3.2.3. Utilization of Health Care and Social Institutions

There are considerable differences among the city quarters regarding children’s utilization of healthcare and social institutions. In quarters with low socioeconomic burden, kindergarten enrolment is at 97.6%, whereas in quarters with high socioeconomic burden, it is merely 80.7%. Particularly concerning is the difference in children having completed the recommended U8 pediatric check-up: 42.9% of children in quarters with high socioeconomic burden had not undergone this check-up, which is in stark contrast to merely 13.2% of children in quarters with low socioeconomic burden who missed it. Figure 3 shows the prevalence of having missed the recommended U8 pediatric check-up.

### 3.3. Regression Results

One of the aims of the SEE is to assess whether a child has attained the developmental skills to start school. Among the variables collected in the SEE, the overall development score is likely the single most important predictor of future academic success. We therefore chose the dichotomous outcome of whether the child had received an overall sub-par developmental score at the time of the SEE for further analysis in the logistic regression.

We first performed a simple logistic regression for the association between the quarter (independent variable) and sub-par development (dependent variable) in the crude model. In the next model (model A), we adjusted for individual BS. Subsequently, we adjusted for the four individual components of the BS in a further model (model B): Mother’s education, mother’s employment, father’s education, and father’s employment.

The unadjusted odds ratio of having a sub-par developmental score was 16.6 (95% CI 10.3–26.6) for quarters with high socioeconomic burden as compared to quarters with low socioeconomic burden. Adjustment for the BS yielded a sizeable reduction in the odds ratio to 8.5 (95% CI 5.2–13.8) for the quarters with high socioeconomic burden as compared to quarters with low socioeconomic burden. Adjustment for parents’ education and employment status in model B produced an odds ratio of 4.9 (95% CI 3.0–8.1) for quarters with high socioeconomic burden in comparison to quarters with low socioeconomic burden.

For the quarters with medium-level socioeconomic burden in comparison to quarters with low socioeconomic burden, the differences between the models were less pronounced: The unadjusted OR for receiving a sub-par result in the development tests in quarters with medium-level compared to low-level socioeconomic burden was 2.8 (95% CI 1.7–4.6). Adjustment for the BS yielded comparatively little change with an OR of 2.4 (95% CI 1.5–3.9). In model B, with the adjustment for the four separate parental characteristics, the OR was 1.9 (95% CI 1.1–3.0).

All adjustments showed a considerable reduction in the odds ratio; nevertheless, the difference between quarters with high socioeconomic burden and quarters with low socioeconomic burden remained sizeable. Full details of the logistic regression can be found in Table 2.

## 4. Discussion

The aim of this paper was to examine whether the socioeconomic differences between quarters of residence are reflected in preschool children’s health.

### 4.1. Residence in Neighborhoods with High Socioeconomic Burden and Perinatal Parameters

We found a higher prevalence of low birthweight and premature birth in the quarters with high socioeconomic burden. These findings are in line with previous research showing higher risks for negative birth outcomes for children born to mothers living in deprived areas [29,30]. The causal pathways of this association are still not fully understood. However, a meta-analysis of studies conducted in the United States proposes that living in a deprived area leads to greater exposure to crime, inadequate nutrition, economic instability, and fewer opportunities for social mobility [31]. In turn, these factors can cause an increase in maternal stress hormones leading directly to decreased blood flow through the placenta and premature birth. Placental hypofusion can lead to negative birth outcomes including low birthweight and stunted intrauterine growth [31]. Moreover, even short times of maternal malnourishment can lead to a deterioration in fetal health [32,33]. Furthermore, previous research suggests that living in deprived areas leads to an increase in perceived stress [19] and is associated with more risky health behaviors. These effects occurred irrespective of individual socioeconomic status [34].

While the structure and demography of our research setting bear some differences to the US urban areas examined in these studies, some of the same causal pathways may still apply to our research setting, since increased stress levels’ association with characteristics of the neighborhood has also been shown in German studies [35].

The observed differences among Halle’s quarters regarding poor birth outcomes are important beyond neonatal health, as low birthweight and premature birth are established risk factors for poor health outcomes in later life, such as diabetes type I and II [36] and cardiovascular diseases [25]. Therefore, it could be assumed that a considerable number of children in Halle’s quarters with high socioeconomic burden are at a higher risk for later disease already at the very start of their lives.

### 4.2. Growing up in a Deprived Environment

Children living in quarters with high socioeconomic burden in Halle have a greater prevalence of being overweight and for sub-par performances in developmental tests. Similarly, childhood deprivation has been linked to a higher prevalence of a variety of poor health and developmental outcomes. These include respiratory illnesses, poor dental health [5], and overweight [37,38].

The underlying pathways leading to the clustering of poor child health have been discussed extensively in the literature. Some of these can likely be applied to our context: Deprived areas are frequently subject to heavy traffic and poor air quality, which can exacerbate respiratory illnesses. This is a significant health threat and has been shown to lead to child death [39]. Furthermore, childhood overweight in deprived neighborhoods has been linked to the limited availability of child-friendly outdoor facilities, such as parks and playgrounds [40], and parents’ perceiving their neighborhood as unsafe and placing restrictions on children’s outside play [41]. It may also be a consequence of inadequate nutrition, as living in a socio-economically deprived environment has been shown to be associated with making high-calorie food choices.

The results of the SEE’s developmental tests showed considerable differences among Halle’s quarters. This supports previous research showing that children from deprived socioeconomic backgrounds have poorer developmental outcomes than the general population [5,42]. In particular, the high levels of sub-par tests of overall development are concerning. We also found high levels of poor fine motor skills in quarters with high socioeconomic burden, which would likely hamper early school education such as learning to write.

The foundation for these deficits may be a suboptimal uterine environment due to malnutrition, which has been linked to children being born with brain defects and an increased risk of neuropsychiatric disorders [43]. Individual low socioeconomic status has been shown to be associated with altered white matter organization [44], as well as suboptimal development of language and executive functions of brain systems [45]. Furthermore, these negative effects can bear lifelong consequences, as childhood poverty has been associated with altered hippocampal and memory functions in adulthood [6].

Even children with higher socioeconomic status living in quarters with high socioeconomic burden may be negatively affected in their developmental outcomes, as living in a poor neighborhood led to altered neural pathways in children, impairing their ability to exert self-control [46].

Another causal pathway could be a lack of learning support in the children’s home and social environment. Reading aloud to children has been shown to have a beneficial influence on cognitive development [47], as does adequate access to appropriate toys [48]. Kindergarten enrolment, which many children in quarters with high socioeconomic burden lack, has also been shown to promote learning and development [49,50,51].

### 4.3. The Potentially Mitigating Role of Social Institutions and Healthcare

Being born into a deprived environment and then growing up in it both carry long-term consequences for health, development, and future prospects [52].

Some factors may be able to mitigate these negative influences. These include good access to healthcare, which would allow early diagnosis of health and developmental issues, and support from social institutions [51]. However, our findings regarding kindergarten enrolment and pediatric check-ups indicate that considerable numbers of children in quarters with high socioeconomic burden do not currently utilize these institutions and therefore cannot benefit from their potential support.

It is likely that some of the poor test results are due to language barriers, as the SEE is performed in German and a considerable number of children in deprived quarters were not born in Germany and may not yet have achieved the same grasp of the language as their German-born peers. For these children, improved access to social institutions such as kindergarten could be particularly beneficial [53]. The high number of children achieving sub-par results in the developmental tests is in particular contrast to the very low number of children actually receiving a referral. This might indicate physicians’ awareness that these results may indeed not be caused by pathology but by not being native speakers of German. However, even if these results are not due to developmental delays, they still show that these children are at a considerable disadvantage, which will likely affect their success in school.

The regression model showed that part of the association between quarters and developmental delays can be explained by social status in the form of BS and its separate components. Previous research has shown that the mother’s characteristics may be particularly important [54].

However, as some difference remained after adjusting for the BS in model A and its individual components in model B, it is likely that living in quarters with high socioeconomic burden constitutes a risk factor in itself. This, in turn, would need to be addressed via community-centered interventions, as has been suggested by previous researchers [55,56,57]. This type of intervention could mitigate poor birth outcomes by improving maternal health and socioeconomic circumstances.

The issue of child poverty and growing urban segregation has attracted considerable public debate in German media. For the specific case of Halle, suggestions have included the introduction of a mandatory limit on child poverty per quarter, in the form of a child poverty quota [58]: Families with children dependent on government handouts should be given the option of moving to a quarter with low child poverty. This would effectively limit the number of families dependent on government aid per quarter and counteract segregation. However, this suggestion has been met with skepticism by the city administration [58]. The city of Halle is situated in the area of the former German Democratic Republic (GDR). Some quarters that have high socioeconomic burden today were once considered the model of socialist urban planning, with relative prosperity and well-connected infrastructure. After German reunification, these quarters underwent a drastic demographic change, due to the emigration of wealthier inhabitants. This has been considered the beginning of social segregation [59]. In recent years, these same quarters have undergone another drastic demographic transformation due to an influx of Syrian refugees. These quarters in particular are considered to be arrival spaces for new refugees, as migrants tended to cluster in quarters with high socioeconomic burden due to lower housing costs [60]. This dynamic has further increased the demographic segregation of the city.

### 4.4. Strengths and Limitations of This Study

The most important limitation of this study is that the financial circumstances of the children’s families were not known. However, a demographic analysis of our study population found very few children with high BS living in quarters with high socioeconomic burden. Even though the BS does not consider a family’s financial situation, a family with low BS is very unlikely to be wealthy. Furthermore, previous research shows that most quarters of Halle are indeed segregated by wealth [14,61].

In the case of single-parent families, the BS doubles the points of the single parent to reflect a hypothetical two-parent household. This method could result in a BS score that overestimates the actual social status of a single-parent family.

Another limitation is that the duration of the children’s residence in Halle’s respective quarters is not known. It is therefore not possible to assess the length of time of their exposure to the neighborhood’s characteristics. Families may have moved to their current residence only recently or have been living there for generations.

Furthermore, there may be relevant individual influences that were not included in the dataset, such as the main language spoken at home, parental age at delivery, and parental health behaviors beyond smoking. Children whose native language is not German may be at a disadvantage when completing the SEE. However, as the child’s knowledge of German is not noted, we have no way to analyze the influence of language skills on the SEE’s results. Similarly, there may be relevant influences due to the quarter’s infrastructure. These influences could affect the relationship between the quarter and the various outcomes; they will be the subject of further research by this working group. The specific causal pathways leading to the results observed in Halle need further investigation in future studies.

## 5. Conclusions

When considering a city or area’s poverty level as a whole, important information remains obscured: While Halle’s overall results still firmly place it within the bottom set in terms of child health, the true extent of the problem becomes hidden by the moderating effect of the average.

Our findings indicate that the city of Halle is segregated not only by wealth but also by health. The roots of this segregation can be found in historic circumstances, but it is exacerbated by current political and economic developments. As many cities in the former Eastern bloc have undergone similar demographic transitions [62], our findings can likely be transferred to other urban contexts, both in Germany and beyond.

Our results show that children in quarters with high socioeconomic burden are faced with concrete developmental and physical disadvantages, which are already manifest at the age of five. In particular, the prevalence of sub-par development is concerning. In combination, these sub-par physical and developmental results indicate that children living in quarters with high socioeconomic burden are at an elevated risk of both later disease and educational failure, even after considering their individual family circumstances. Regardless of the origins of these deficits, they put children at an educational disadvantage even before their first day of school. A possible consequence is a lack of social participation and transmission of poverty and poor health to future generations [63].

Every fifth child in Germany is currently at risk of poverty, with less than 60% of the German average household income. There has been a sharp increase in recent years in the number and proportion of children living in households that are dependent on government subsidies for survival. This has been exacerbated by the recent influx of Ukrainian child refugees [10]. There also is a sharp increase in the cost of living throughout Europe. These developments may further exacerbate socioeconomic segregation in the future. This makes the implementation of interventions to tackle these inequalities a social and political imperative. To gain a better understanding of these inequalities, we recommend that future research include data concerning family income and the child’s knowledge of German. As segregation and inequalities may lead to prejudice and discrimination, pertinent information regarding these experiences should also be included in future data collection.

As the quarter showed a considerable association with all risk factors examined, interventions should be implemented at a local level, improving access to community resources [57]. However, improving community resources and trying to mitigate the medical and social effects of deprivation is only one part of the solution [55]. Ultimately, the priority should be to address the socioeconomic and educational inequality of the parents as the roots of childhood poverty, as well as the lack of social cohesion.

## Figures and Tables

**Figure 1 children-10-00865-f001:**
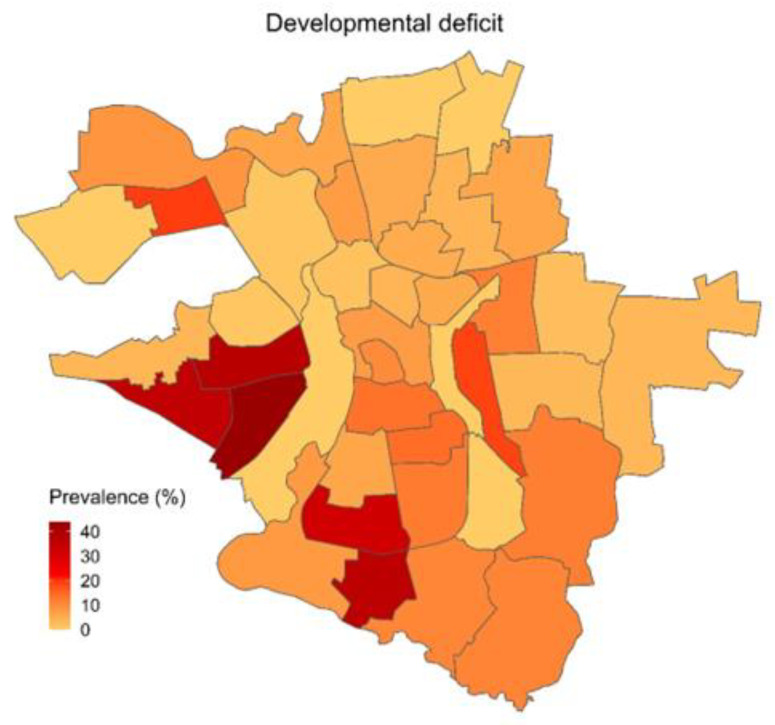
Prevalence of an overall developmental score below the cut-off value.

**Figure 2 children-10-00865-f002:**
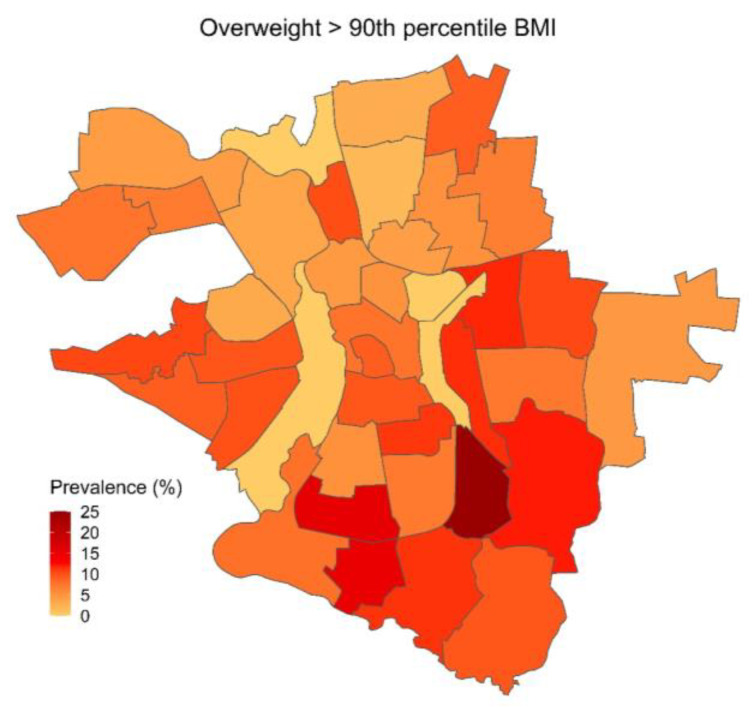
Prevalence of overweight at the time of the SEE across Halle.

**Figure 3 children-10-00865-f003:**
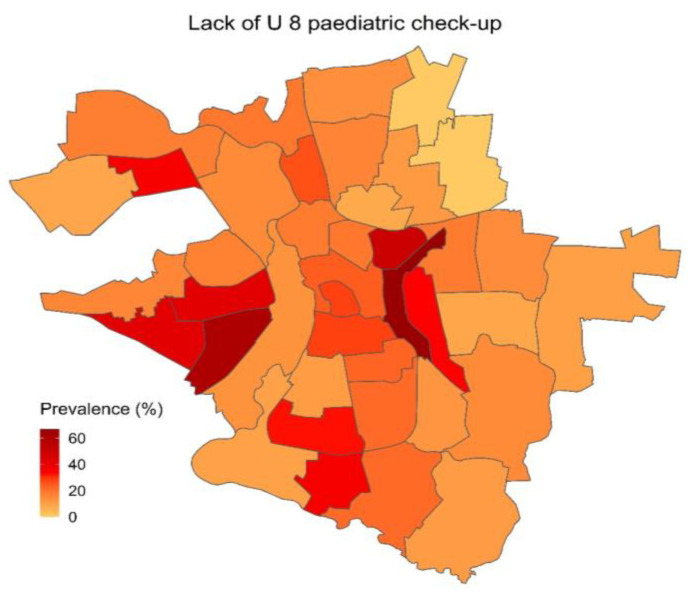
Prevalence of children not having undergone the recommended *U8* pediatric check-up.

**Table 1 children-10-00865-t001:** Sociodemographic characteristics are distributed unequally across the quarters.

	Low Socioeconomic Burden Quarters (n = 552)		Medium Socioeconomic Burden Quarters (n = 4316)		High Socioeconomic Burden Quarters (n = 2686)		City of Halle (n = 8417)	
	%	N	%	N	%	N	%	N
Female sex	49.5	273	47.8	2061	49.2	1321	48.5	4081
High BS	54.5	301	41.5	1789	8.5	228	28.9	2435
Medium BS	16.5	91	23.0	993	24.5	659	21.9	1846
Low BS	2	11	7.4	318	31.3	841	14.7	1238
Unknown BS	25.0	149	28.2	1216	35.7	958	34.4	2898
Single parent households	8.9	48	17.0	735	32.4	869	21.77	1832
Both parents in household	84.8	468	73.8	3187	53.5	1438	67.3	5668
Attending Kindergarten	97.6	539	94.9	4096	80.7	2168	90.4	807
Smoking household	18.1	100	30.6	1319	49.9	1341	36.2	3048
Child born in Germany	96.7	534	92.4	3986	75.7	2032	86.7	7296
Born prematurely < 37 weeks	6.9	36	9.1	345	11.2	210	9.4	646
Low birthweight < 2500 g	4.9	26	6.7	257	9.9	188	7.3	514
Currently overweight BMI > 90th percentile	5.3	29	8.0	341	11.3	295	8.9	731
No U 8 health screening	13.2	73	21.8	942	42.6	1151	28.6	2406
Sub-par grammar	1.8	10	8.5	367	35.6	957	17.4	1460
Sub-par fine motor skills	3.8	21	6.2	266	17.2	461	9.7	816
Sub-par gross motor skills	2.5	14	2.3	97	7.7	206	4.2	357
Sub-par cognitive development	1.5	8	3.9	170	17.2	462	8.2	691
Sub-par overall development	3.3	18	8.7	375	35.8	962	17.7	1490
New specialist referral	7.8	43	8.4	364	16.2	434	11.2	943

**Table 2 children-10-00865-t002:** Influences on sub-par development tests.

	Crude Model OR, 95% CI	Model A * OR, 95% CI	Model B ** OR, 95% CI
Medium socioeconomic burden quarters vs. low socioeconomic burden quarters	2.8 (1.7–4.6)	2.4 (1.5–3.9)	1.9 (1.1–3.0)
High socioeconomic burden quarters vs low socioeconomic burden quarters	16.5 (10.3–26.6)	8.5 (5.2–13.8)	4.9 (3.0–8.1)
Unknown BS vs. High BS		5.9 (4.6–7.6)	
Medium BS vs. High BS		2.4 (1.8–3.1)	
Low BS vs. High BS		9.9 (7.6–12.9)	
Father: Education level unknown vs. more than 10 years of education			2.0 (1.4–2.7)
Father: 10 years of education vs. more than 10 years of education			1.4 (1.1–1.7)
Father: Less than 10 years of education vs. more than 10 years of education			2.3 (1.8–2.9)
Mother: Education level unknown vs. more than 10 years of education			1.9 (1.4–2.7)
Mother: 10 years of education vs. more than 10 years of education			1.0 (0.8–1.2)
Mother: Less than 10 years of education vs. more than 10 years of education			1.9 (1.5–2.4)
Father: Employment status unknown vs. employed			0.9 (0.7–1.2)
Father: Unemployed vs. employed			1.5 (1.3–1.9)
Mother: Employment status unknown vs. employed			3.6 (2.6–5.0)
Mother: Unemployed vs. employed			3.8 (3.2–4.5)

* Crude Model + BS. ** Crude + mother’s and father’s employment and education.

## Data Availability

The data that support the findings of this study are available from the Public Health Department, City of Halle (Saale), Germany (address: Fachbereich Gesundheit, Niemeyerstraße 1, 06110 Halle (Saale), Germany), but restrictions apply to the availability of these data, which were used under license for the current study and therefore are not publicly available. Data are, however, available from the authors upon reasonable request and with permission of the Public Health Department, City of Halle (Saale), Germany.

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
