# Peer review of "Segregated by Wealth, Health, and Development: An Analysis of Pre-School Child Health in a Medium-Sized German City"

_children, 2023, doi:10.3390/children10050865_

Round 1

Reviewer 1 Report

The study presents some corelation between socioeconomic factors with some type of exam on five years old children in specific city in Germany. The collected data in the exam were tried to be correlated with socioeconomic quarters of the city.  Yet the explanations about these terms and exams are not enough to understand and forms some contreversial ideas. Such as what is U-Heft. It should be ezplained. Also, There is ambiquition in usage of quarters in the text. It should be resolved. the Brandenburger Sozialindex (BS) should be explained in the manuscript also. "high or low socioeconomic burden" the usage burden with high or low socioeconomic terms feels some contreversial ideas.  

As I understood there is huge amount of collected data but not studied specifically. The study is very broad from education to medicine yet results and conclusions are not compatible. Instead, some specif points should be studied with some correlations. 

As a result the manuscript has written in hazy manner the idea is written in random manner that what it convesy is difficult to understand.

Reviewer 2 Report

This manuscript investigates the disparity problems across quarters in a German city. The disparity is manifested by the prevalence rate of sub-par development, health screening, birthweight, etc. While the topic is important, there are serious flaws in the empirical analyses. The details are as follows.

1.     The method is not clearly explained; thus, the results are perplexing. The logistic regressions investigate the association between the quarter and sub-par development. It is not understood how the data are coded or stored for any of the variables used.

2.     The data needs more description. Many of the variables are not self-explanatory. For example, how the socioeconomic burden is defined to be low, medium, and high; the BS index is not familiar to many readers and needs further explanation.

3.     The logistic regression in its current form is pre-mature. A lot more explanatory variables could have been included to hold the model more accountable, such as gender ratio, ethnicity ratio, income levels, education ratio, selected disease prevalence rate, and the number of schools, hospitals or clinics in each quarter. The characteristics of the quarters might not be readily available from just one data set. But there should be other sources to obtain this information.

4.     The presentation of the regression result is not appropriate. All the coefficients of the independent variable, at least the key independent variable, should be reported. 

Reviewer 3 Report

The article is interesting and well presented, with a correct statistic analysis and good results. Conversely, the minimal area where it was produced could hinder wide dissemination and interest many readers; that's the main concern.

Reviewer 4 Report

The authors examine the comparative  health status of children in a German City's particular areas or quarters which are distinguished from each other by  socioeconomic differences using data collected using the School Entry Examination (SEE).  As perhaps expected, the findings indicate that socioeconomic differences matter as indicated by quarters with high as opposed to low economic burden were found to have children with higher incidence cognitive development deficiencies and sub-par development as well as being at greater risk for developing health problems and diseases later in life. This study is significant in that it provides further evidence for the impact of existing socioeconomic disparities between different areas of  a city with lasting consequences for its youngest citizens, even when parents' education and employment status are controlled in the analysis. This drilling down and focus on explaining the factors for the differences observed will contribute to the literature.

There are some parts of the manuscript that can be enhanced. These are the following:

1.  A general description of geographic description and demographic characteristics of the  German City population from which the data was selected  in addition to poverty will be helpful to better contextualize the results reported later in the manuscript.  

2. Introduction section: lines 54-56.  Provide additional language and citations to indicate that this has been found to be a case throughout the world, and a significant problem in other cities in western countries. For instance,  in the U.S., a particular emphasis has been on from a public health perspective, research on social determinants of health (SDOH) and how they have been show to impact individual wellbeing.  Although some variation of this is mentioned in the discussion section, it would be important to include this early on so as to provide greater context for later discussion of implications of the results and recommendations.

3. Materials and methods section: lines 130-138. Provide a fuller description of the Brandenburger Sozialindex (BS) for readers not familiar with this tool.

4. Materials and methods section: lines 140-141. Please  explain rationale for not using quarters with less than 30 children?

5. Materials and methods section: lines 141-142, the authors mention that they divided quarters into High, Medium, and Low building on pre-existing research, the specifics of what constitutes High, Medium, and Low should be stated in the narrative or indicated in the table or stated as a footnote/endnote.

6.  In the discussion section, lines 335-338, the authors state the SEE is performed in German, is this a study limitation, in regard to measurement tool used?  Relatedly, the general  limitations of  using secondary data, data not originally collected for research would be important to note. 

7. In the discussion section, the authors should elaborate further on the historical and current structural arrangements and neighborhood characteristics and conditions that are the basis for the perpetuation of the child poverty and urban segregation that are contributing to children's ill health and pose future risk to their wellbeing.  Although not part of the study and analysis, the authors' description of residents living in high socioeconomic burden areas suggests  whether discrimination, prejudice, and bias should be considered as part of the discussion; and whether  attitudes towards settlement of non- German born are playing a  role in impeding any efforts to change such as the one suggested  in lines 356 to 363.   What recommendations would they have for research and interventions to address these issues?

Round 2

Reviewer 2 Report

The authors have made clear modifications/explanations to the suggestions I gave.  Even though some of the responses are not exactly satisfactory, it might have reached the standards of the journal. I have no further suggestions.